# Three-dimensional wide-field fluorescence microscopy for transcranial mapping of cortical microcirculation

Quanyu Zhou [1,2,5], Zhenyue Chen[1,2,5], Yu-Hang Liu[1,2], Mohamad El Amki[3,4], Chaim Glück[1,4], Jeanne Droux[3,4], Michael Reiss[1,2], Bruno Weber [1,4], Susanne Wegener[3,4] & Daniel Razansky [1,2,4] ✉

Wide-field fluorescence imaging is an indispensable tool for studying large-scale biodynamics. Limited space-bandwidth product and strong light diffusion make conventional implementations incapable of high-resolution mapping of fluorescence biodistribution in three dimensions. We introduce a volumetric wide-field fluorescence microscopy based on optical astigmatism combined with fluorescence source localization, covering 5.6×5.6×0.6 mm³ imaging volume. Two alternative configurations are proposed exploiting multifocal illumination or sparse localization of point emitters, which are herein seamlessly integrated in one system. We demonstrate real-time volumetric mapping of the murine cortical microcirculation at capillary resolution without employing cranial windows, thus simultaneously delivering quantitative perfusion information across both brain hemispheres. Morphological and functional changes of cerebral vascular networks are further investigated after an acute ischemic stroke, enabling cortex-wide observation of concurrent collateral recruitment events occurring on a sub-second scale. The reported technique thus offers a wealth of unmatched possibilities for non- or minimally invasive imaging of biodynamics across scales.

Intravital fluorescence imaging is widely employed in biological research and medical diagnosis owing to its excellent molecular specificity and sensitivity. Among the multitude embodiments of this technique, wide-field fluorescence imaging can achieve kilohertz (kHz) frame rates over large (centimeter-scale) field-of-view (FOV). Yet, lack of depth information imposes significant limitations for many applications requiring optical sectioning and quantification abilities[1]. Fluorescence molecular tomography allows for detection and quantification of fluorescence signals in three dimensions (3D) in deep tissues. However, it involves protracted tomographic scans and suffers from poor spatial resolution, typically 1 mm or worse in living tissues[2]. In contrast, scanning intravital confocal[3] and multi-photon[4–6]

microscopy can achieve diffraction-limited optical resolution in 3D at the expense of restricted FOV and/or slow volume rates, making these techniques suboptimal for studying fast biodynamics on a large (e.g., whole-cortex) scale. Although parallel excitation/detection strategies such as light-sheet microscopy[7,8], spinning disk confocal microscopy[9] and structured illumination microscopy[10–12] can greatly boost the 2D frame rate while maintaining relatively large FOV, the depth resolving ability relies on mechanical axial scanning of the sample/objective, thus inevitably rendering a low volume rate when imaging of centimeter-scale FOVs. Conversely, several miniature epifluorescence microscope designs have recently been introduced for wide-field imaging in real-time[13–17]. Those techniques are again limited to planar

¹Institute of Pharmacology and Toxicology, Faculty of Medicine, University of Zurich, Zurich, Switzerland. ²Institute for Biomedical Engineering, Department of Information Technology and Electrical Engineering, ETH Zurich, Zurich, Switzerland. ³Department of Neurology, University Hospital and University of Zurich, Zurich, Switzerland. ⁴Zurich Neuroscience Center, Zurich, Switzerland. ⁵These authors contributed equally: Quanyu Zhou, Zhenyue Chen. ✉ e-mail: daniel.razansky@uzh.ch

(2D) imaging with signals integrated along the depth, thus impacting quantification. Furthermore, the skull presents a significant barrier for high resolution imaging of the murine brain. Thus, the existing optical microscopy approaches commonly involve high degree of invasivity, such as installation of cranial windows, skull thinning or invasive implantations.

Several advanced methods were proposed to empower fluorescence microscopy with fast volumetric sectioning capacity. A straightforward solution is adding an electrically[18] or acoustically[19] tunable lens before the objective that can provide axial scanning rate of several kHz at the expense of limited effective numerical aperture. An aberration-free remote focusing was introduced through matching the objective with a second objective combined with an actuated tilted/stepped mirror while maintaining the diffraction-limited performance[20,21]. An alternative approach is based on astigmatism, which was firstly applied to super-resolution microscopy (e.g., 3D stochastic optical reconstruction microscopy) by inserting a weakly focusing cylindrical lens in the imaging light path[22]. Besides cylindrical lenses, point spread function (PSF)-engineered phase masks, such as double-helix[23] and tetrapod[24], were employed to encode depth information into the PSF shape. A similar strategy is called "bi-plane" detection through creating two slightly separated focal planes with two detectors[25]. Elongated "V-shape" dual-beam illumination was also reported to retrieve z coordinate from the displacement of "image pairs"[26]. However, most of these methods serve as the add-on module for conventional point scanning microscopy thus are anyway afflicted with relatively slow imaging speed.

All in all, real-time 3D fluorescence imaging with high spatial resolution over large FOV remains an unmet need. Here we show that astigmatism-based imaging of localized fluorescence sources created via multifocal illumination or sparse distribution of fluorescence emitters results in volumetric wide-field fluorescence microscopy with high spatio-temporal resolution. We then demonstrate the preclinical in vivo application of the newly developed technique for investigating collateral vessel recruitment in ischemic stroke, which is known to crucially affect infarct severity, treatment efficacy and recovery[27,28].

## Results

### Integrated 3D wide-field fluorescence microscopy system and characterization

The two proposed implementations of the 3D wide-field fluorescence microscopy share the same detection path while differing in the illumination path (Fig. 1a). For the sparse localization (SL) approach, the laser beam is coupled into a commercial fiber bundle to provide epi-illumination. The multifocal illumination (MI) approach uses instead a structured laser beam projected as a "lattice" pattern on sample's surface. Synchronization of acousto-optical deflector (AOD) and a high-speed camera ensures fast scanning with frame rate up to 40 Hz across a $5.6 \times 5.6$ mm² lateral FOV. A custom-made cylindrical lens with an extra-long 8200 mm focal length was inserted between the objective and tube lens to create two slightly separated focal planes encoding an effective depth range of ~600 μm. Note that the focal length of the cylindrical lens was chosen based on the magnification power of the microscope, which eventually determined the effective depth range. After calibration, the depth information of fluorescent emitters is calculated according to the elongated PSF (Fig. 1a, right panel). More details on the experimental set-up are provided in the online methods.

The lateral resolution of the system was characterized with randomly distributed fluorescence beads on a flat microscope slide. According to the Gaussian fitted line profile along a single bead, the system has a lateral resolution of 12.6 μm, as determined by the excitation spot size (Fig. 1b). To establish the relationship between the PSF shape and depth information, a 3D stack of images of the fluorescent microscope slide was acquired from different axial locations.

Subsequently, the fitted curve of PSF ellipticity versus depth served for calibration. However, due to aberrations from the imaging system, the calibration curve exhibits large fluctuations, especially for large FOV (Fig. 1c). To improve on accuracy of the depth estimations, spatially variant "PSF shape-to-depth" calibration files were generated by dividing the entire FOV into 25 sub-regions, each having a local calibration file (Fig. 1c). Consequently, the calibration curve fluctuations in the sub-regions were greatly suppressed within the entire depth range of ~600 μm. Based on the depth calibration files, the axial resolution of the system was characterized by imaging a tilted fluorescent slide that provided a continuously changing depth (Fig. 1d). With the proposed method, the approximate linear relationship between the fitted depth and lateral displacement along the depth gradient corroborates accurate depth estimations (Fig. 1e). The accuracy of depth estimation was characterized by the averaged standard deviation (SD) across the entire depth range, which was calculated to be 5.5 μm. The axial resolving accuracy was further validated through imaging fluorescence beads embedded in the UV-curing glue. By shifting the phantom along z axis with 10 μm step size using a motorized stage, the estimated depth exhibits an approximately linear relationship versus axial displacement of the beads (Supplementary Fig. 1). The actual 3D imaging performance of the system was validated by imaging two crossing microfluidic channels separated by approximately 260 μm in the axial dimension. The experimentally estimated depth difference amounted to $239.2 \pm 31.3$ μm (Fig. 1f).

### In vivo transcranial mapping of cerebral microcirculation

The in vivo 3D imaging capability was demonstrated by transcranial mapping of the cerebral microcirculation in mice with the scalp removed. For the SL imaging, fluorescent beads were injected intravenously into a nude mouse. A sequence of wide-field images was recorded at 200 frames per second (fps) for a total duration of 3.5 min post injection. The individual flowing beads were continuously localized with the high-resolution depth-resolved brain microcirculation map rendered by tracking their trajectories in all three dimensions (Fig. 2a and Supplementary Movie 1). As expected, the map exhibits an intrinsic depth gradient owing to the curvature of the mouse skull and brain. Furthermore, the calvarian and cerebral vessels are distinguishable based on the color assigned to their relative depths. The ~150 μm depth variance of skull vessels is in good concordance with previously reported mouse skull thickness at this age[29]. This is further corroborated by the blood velocity map (Fig. 2b), an intrinsic byproduct extracted during the localization process, where the skull vessels exhibit significantly lower velocity as compared to the cerebral vasculature. Max-intensity-projection (MIP) of the ROI indicated by the red square (Fig. 2a) along with its side view of selected slices are shown in Fig. 2c. The practical axial resolution was determined by the PSF of a capillary along depth direction, which was about 16.6 μm (Fig. 2c, bottom). The good correspondence between the depth and velocity maps is further manifested in exquisite detail in the expanded views shown in Fig. 2d, e. Under ideal conditions, the lateral resolution of the SL approach is determined by the photon budget and pixel size. A line profile plotted along the white line shown in Fig. 2d reveals that two neighboring features with lateral displacement of 7.4 μm could be distinguished unambiguously (Fig. 2f), i.e., closely resembling the effective pixel size given its 11 μm × 11 μm camera pixel size at 1.5× magnification power. We validated the velocity measurement with the proposed method using line-scanning two-photon microscopy (2PM), which was performed in the same mouse after cranial window implementation (Fig. 2g, h). A good correspondence was observed between the two methods (Fig. 2i, j) with SL enabling transcranial measurements and superior scalability for parallel velocity measurement across large FOV.

Although the SL method is able to achieve high spatial resolution, its temporal resolution is limited since thousands of consecutive

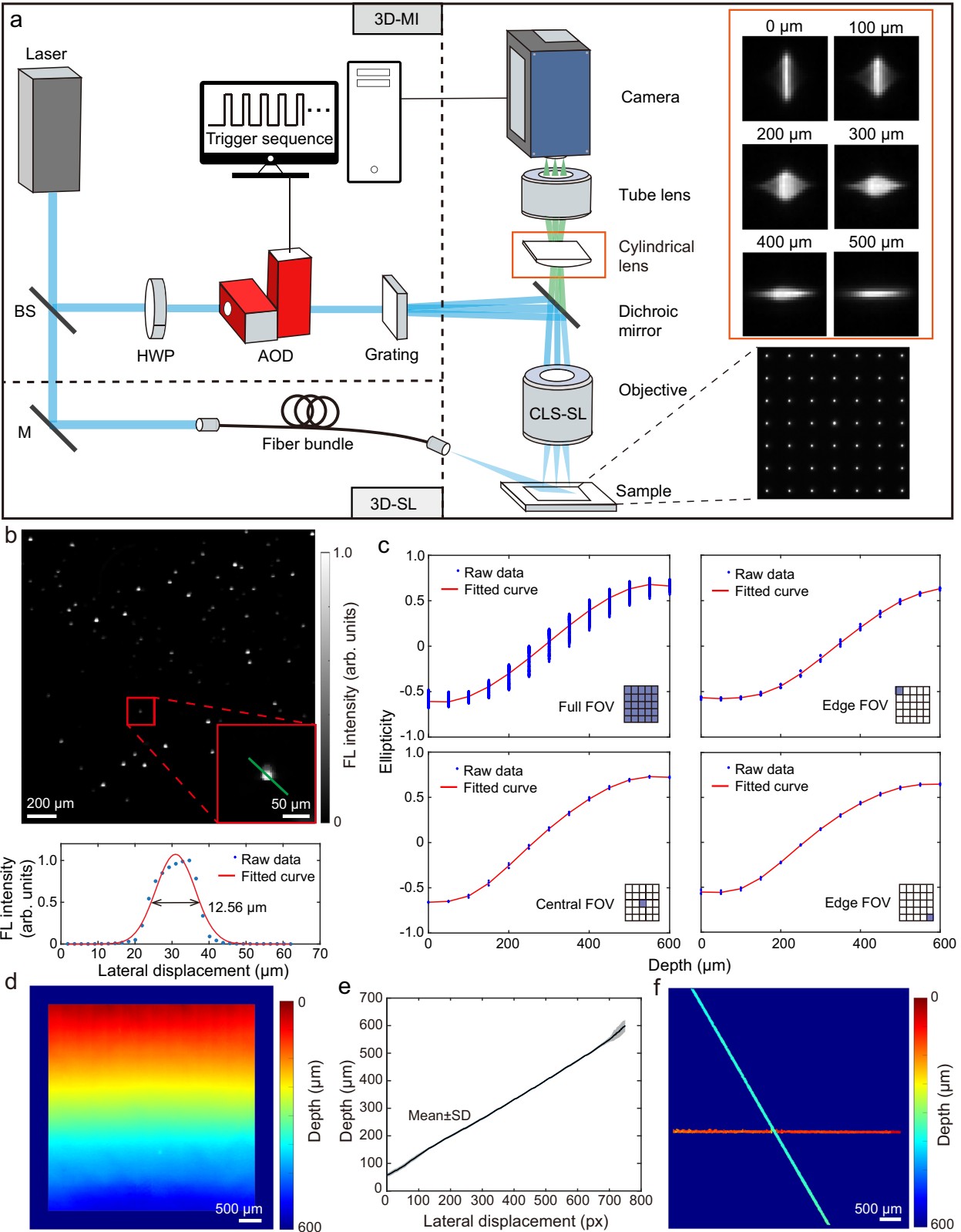

**Fig. 1 | The integrated volumetric wide-field fluorescence microscopy system based on optical astigmatism and its characterization. a** Schematic of the two spatial localization approaches, which are based on either multifocal illumination (MI) or sparse localization (SL) of individual fluorescence emitters. **b** Lateral resolution characterization by calculating the full-width-at-half-maximum (FWHM) of the intensity profile of fluorescent beads after Gaussian fitting along the green line. **c** Representative calibration curves of the PSF ellipticity as a function of depth across the whole FOV (289 illumination spots) and corresponding subregions where 5 illumination spots were selected. **d** 3D fluorescence image of a tilted microscope slide with linear depth gradient along axial direction. **e** Averaged line profile along axial direction across the FOV. Data are presented as mean ± SD. **f** 3D fluorescence image of two crossing tubings separated by approximately 260 μm. BS beamsplitter, M mirror, AOD acousto-optical deflector, HWP half-wave plate, FL fluorescence, FOV field of view, SD standard deviation.

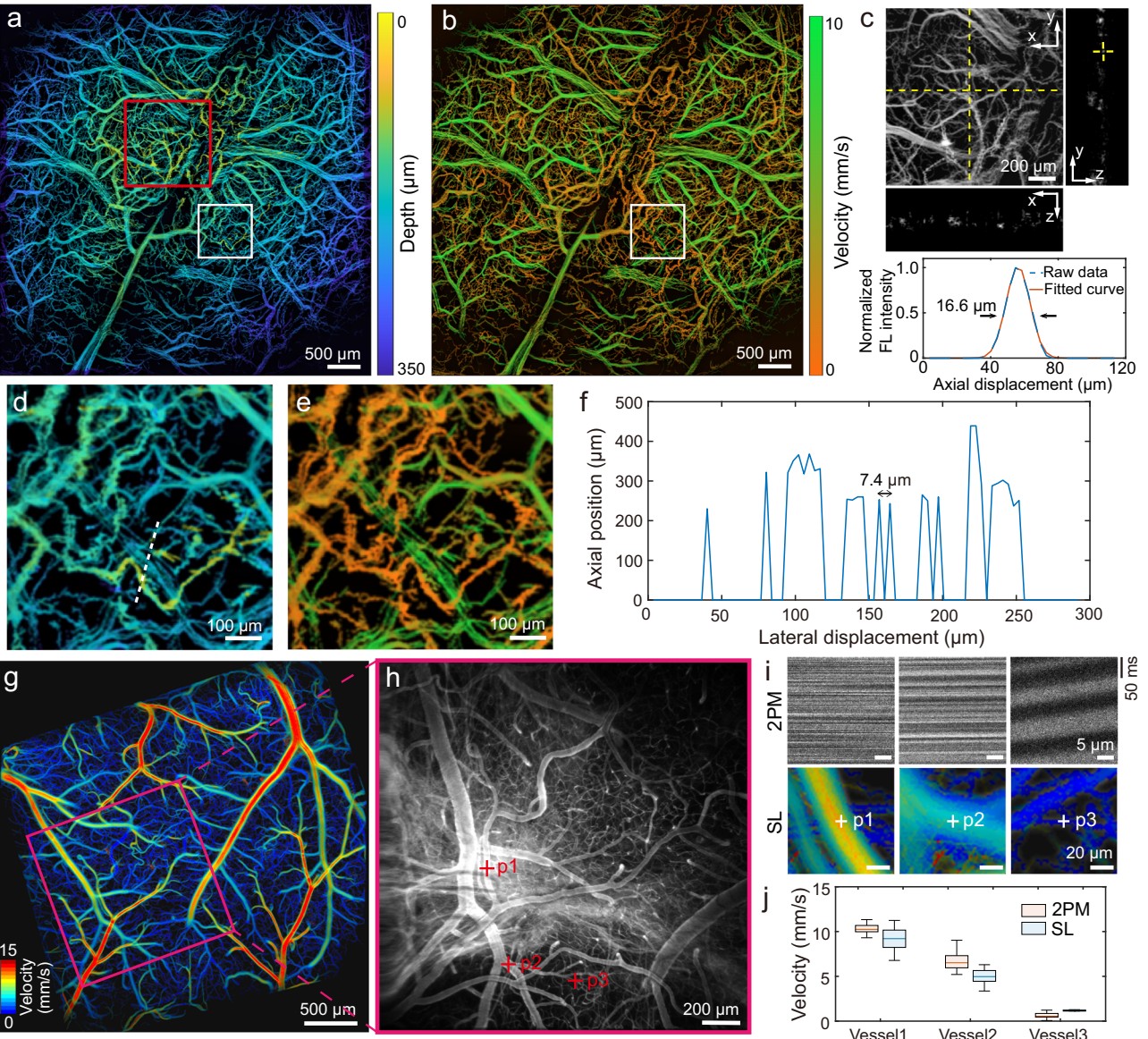

**Fig. 2 | Imaging of microcirculation in murine brain and skull with depth-resolved astigmatism-based sparse localization (SL) method. a** Color-encoded microvascular depth map. **b** The corresponding color-encoded blood velocity map. **c** Enlarged view of the red square ROI in **a** with yellow dashed line indicates where x-z and y-z cross-section views are sliced. The line profile in the z direction is plotted with raw data fitted Gaussian function at the bottom. **d**, **e** Expanded views of the white square ROIs indicated in **a** and **b**. **f** Signal profile along the white dashed line indicated in **d**. **g** Color-encoded velocity map of the mouse with cranial window reconstructed from SL method. **h** Intensity map of the purple square ROI indicated in **g** captured with two-photon microscopy (2PM). **i** Kymographs of red blood cells flow and corresponding velocity map of three selected vessels (marked with red crossings in **h**) captured with 2PM and SL, respectively. **j** Measured velocity with 2PM ($n = 166$) and SL method ($n = 547$, 526, and 8 for vessel 1, vessel 2, and vessel3, respectively) of selected vessels. In the boxplot, minima and maxima are shown as the bounds of whiskers whereas median value, 25th percentile and 75th percentile are shown as the middle, top and bottom lines of the box. Representative data from one mouse are shown.

frames are processed to render a single high-quality compounded image. An additional prerequisite is the sparse distribution of single fluorescent emitters in the blood stream to avoid PSF overlapping, which establishes an optimal concentration range of the injected contrast agent. In general, quality of the SL image depends on number of the recorded fluorescent emitter sources, which can be increased by either higher density of flowing beads, which may compromise localization accuracy, and/or longer recording duration (more frames). Comparison of SL images reconstructed with different recording durations at the given 200 Hz camera frame rate is shown in Supplementary Fig. 2. In contrast, for the MI method the image acquisition is facilitated by exploiting multifocal illumination and fast scanning scheme. To showcase the imaging speed advantages of the MI scheme, we recorded mouse brain perfusion dynamics with Cy5.5 fluorescent dye. Compared to previously reported planar (2D) MI imaging strategies[11,12], a lower SNR was attained due to the enlarged PSF. Therefore, the AOD scanning frequency and high-speed camera frame rate were set to 2.25 kHz while 15 × 15 scanning positions were used as a compromise between temporal resolution and SNR, resulting in 10 Hz effective volume rate for the 3D compounded images. Time-lapse depth maps (Fig. 3a) depict the vascular perfusion sequence with the deep cortical vessels (greenish colors) pronounced before 4 s while more superficially located (yellowish) vessels appearing after 5 s. Superimposition (maximum projection) of the time-lapse image sequence reveals the depth difference between typical skull and brain vessels (Fig. 3b). This result is also in congruence with previous reports

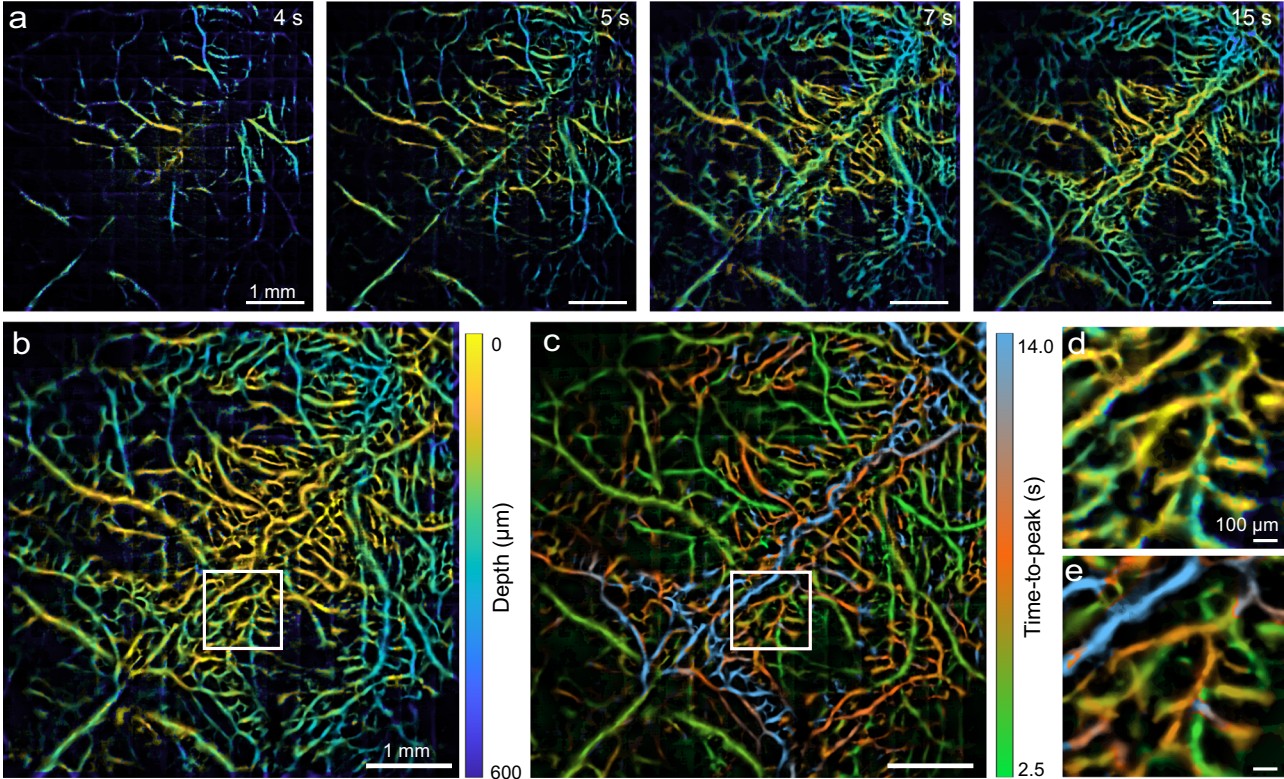

**Fig. 3 | Real-time volumetric microcirculation mapping with astigmatism-based multifocal illumination (MI) method following injection of Cy5.5 fluorescent dye. a** Time-lapse images acquired at different time points post dye injection. **b** Superimposition (maximal projection) of the time-lapse depth maps. **c** Time-to-peak (TTP) map depicting the perfusion dynamics. **d, e** Expanded views of the ROI indicated by the white squares in **b** and **c**. Representative data from one mouse are shown.

showing that brain vessels are the first to be perfused upon intravenous injection of contrast agents[30]. The perfusion dynamics is further mapped according to the time-to-peak (TTP) constant (Fig. 3c), manifesting a high level of correlation with the depth map in Fig. 3b (see also the expended views in Fig. 3d, e).

**Investigation of cerebral perfusion change in the mouse brain post ischemic stroke**

The high volumetric imaging rate and capillary level resolution across centimeter-scale FOV make the MI technique ideal for investigating cerebral hemodynamic changes in healthy and diseased conditions such as ischemic stroke. The depth map and TTP map of a C57BL/6J mouse post-ischemia were obtained by injecting Cy5.5 intravenously (Fig. 4a,b). For comparison, laser speckle contrast imaging (LSCI) of the same mouse was also performed (Fig. 4c). To evaluate the morphological changes of the vascular network after middle cerebral artery (MCA) occlusion, three ROI pairs were selected from both contralateral (healthy) and ipsilateral (infarct) sides (Fig. 4a). Zoom-in views of representative ROIs corroborate the superior spatial resolving power of the MI method compared to LSCI (Fig. 4d). Automatic vessel segmentation and analysis were performed to quantify the vessel fill fraction and vessel branch numbers within the selected ROIs (Fig. 4e). The ipsilateral side displays an evident reduction in both readings (paired-sample t-test, two-sided, $p = 0.09$ for fill fraction and $p = 0.07$ for vessel branch number). Although the difference here is not statistically significant, the corresponding effective size calculated from Cohen's d for paired samples is 1.84 and 2.00, which constitutes a significant difference between groups. In addition to the structural changes, temporal fluorescence perfusion patterns of ROIs ($5 \times 5$ pixels) located at MCA and anterior cerebral artery (ACA) in both hemispheres were extracted from the time-lapse image stack (Fig. 4f).

Statistical analysis on TTP reveals that MCAs on the ipsilateral side have a higher mean and SD value compared to MCAs on the contralateral side, while ACAs were also affected by the infarct with a higher variation of TTP (Fig. 4g). Interestingly, the 100 ms effective temporal resolution of the method allowed the real-time transcranial cortex-wide observation of fast collateral recruitment[28] through additional blood flow originating from the adjacent ACA passed through collateral vessels to feed the obstructed MCA (Fig. 4h and Supplementary Movie 2). Since such observations are hindered with other microscopic tools due to the limited spatiotemporal resolution and FOV, previous studies have chiefly focused on static recording of vessel density, diameter and tortuosity changes in collateral vessel developments within a time window of hours to days post stroke[31]. Owing to the high volumetric rate and the scattering suppression inherent in the MI imaging strategy, the fast collateral recruitment events could be examined both spatially and temporally by simultaneously providing quantitative perfusion information across both brain hemispheres.

## Discussion

This work exploits optical astigmatism for realizing depth-resolved wide-field fluorescence microscopy with localized fluorescence sources, which can be generated via either multifocal illumination (MI) or sparse localization (SL) of point fluorescence emitters. While the SL approach can achieve higher spatial resolution only limited by the source localization accuracy, it has an inferior temporal resolution since thousands of frames are required for rendering a high-resolution compounded image. In contrast, the MI method only requires a quick raster scan across the imaged FOV, being more suitable for studying rapid bio-dynamics on a millisecond scale, such as cortical perfusion. More importantly, unlike single-emitter localization-based methods,

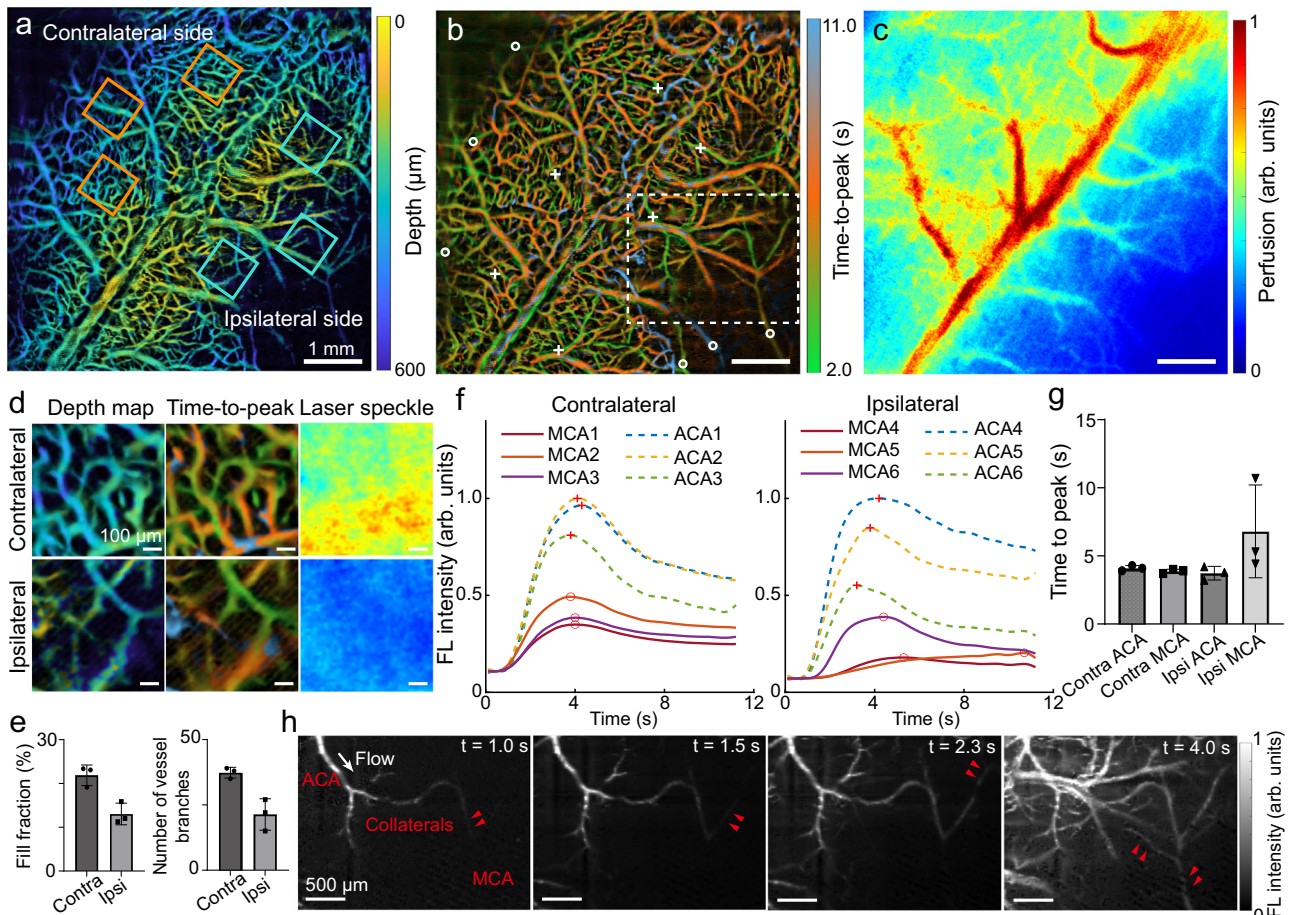

**Fig. 4 | Mapping of collateral recruitment in murine brain post ischemic stroke.**
**a–c** Comparisons of color-encoded depth, TTP map captured with 3D astigmatism-based MI method versus the planar (2D) cortical perfusion map captured with laser speckle contrast imaging. Three pairs of ROIs were selected symmetrically (based on the superior sagittal sinus) in the contralateral (healthy) and ipsilateral (infarct) side indicated by solid line squares. **d** Expanded views of depth, time-to-peak and perfusion image from one of the selected ROI pair. **e** Quantitative comparison of fill fraction and vessel branch number in the selected ROI pairs (*n* = 3) indicated by mean ± SD (paired-sample t-test, two-sided). **f** Temporal perfusion profiles of selected regions situated at ACAs (indicated by white crosses) and MCAs (indicated by white circles) in both hemispheres. **g** Statistical analysis of time-to-peak value (*n* = 3). Data are presented as mean ± SD. **h** Representative time-lapse images acquired by the MI method depicting the additional blood flow compensation from the adjacent ACA to occluded MCA through the bypass collateral vessels. The experiment was repeated independently in 3 mice with similar results.

MI does not impose strict requirements on the type/size of contrast agent nor its in vivo concentration since the cross talk between neighboring emitters is avoided by the sparse grid illumination pattern, hence allowing for a broader applicability of the method.

The axial (depth) resolution in optical astigmatism is determined by the ellipticity of PSF and its projection to the depth look-up table. As a result, the axial resolution is ultimately limited by the image SNR. Axial localization inaccuracies may arise when different emitters located at different depths share the same lateral coordinates. However, our technique greatly mitigates this challenge since the SL approach is based on a high level of spatial sparsity of fluorescent emitters while the MI approach exploits the temporal sparsity owing to the perfusion dynamics. While spatio-temporal sparsity of fluorophores lies at the core of the method's performance, it can potentially be applied to a diverse range of samples since many dynamic processes in the brain are characterized by either spatial and/or temporal sparsity. This applies not only to hemodynamics and blood circulation but also to neural activity and signaling. For instance, with the average rate of neuronal spiking of 5 Hz, over 90% of neurons in some brain areas remain effectively silent, i.e. they spike every ten seconds or more[32,33]. Thus, if the MI method is about to be applied for imaging of neuronal activity e.g. with the help of calcium-sensitive fluorescence dyes or proteins[34,35], significant

temporal variations are only expected to simultaneously occur in a very small portion of voxels.

Note that a number of alternative methods exist to extract the PSF shape (e.g., 2D elliptical gaussian fitting[22] or phasor-based method[36]) where the depth look-up table is constructed based on the axis difference or quotient of two axes (defined as ellipticity) of this asymmetric PSF. However, for in vivo study, the PSF from an ideal point source deep in biological tissue is inevitably blurred due to random photon scattering. Since ellipticity exhibited the highest robustness to the changes in scattering properties compared to other methods (Supplementary Note 1 and Supplementary Fig. 3), it was used to build the calibration curve linking the astigmatic PSF to depth. Intuitively, scattering causes a similar blurring effect to both axes of the PSF. The ellipticity metric is able to efficiently counteract the scattering-induced blurring by considering the quotient instead of relying on the single-axis length. As an additional measure, a local calibration curve (look-up table) collected from targets in scattering media can be incorporated to further alleviate the scattering influence.

It is worth mentioning that other PSF engineering methods exist to encode the depth information in optical imaging[37]. Among them, the Tetrapods[24] and double-helix PSF[23] are two promising strategies that have been used in super-resolution localization microscopy. However, when imaging opaque tissues with a large FOV

(e.g., >1 mm), the latter methods suffer from significant field-dependent aberrations[38]. Additionally, implementing multifocal illumination with Tetrapods and double-helix PSF methods is challenging as additional spacing is required to separate point-pairs on the detector, which in turn restricts the diameter and structure of the phase mask, resulting in a low SNR and temporal resolution. In contrast, the astigmatic PSF can be easily realized by inserting a low power cylindrical lens between the objective and the tube lens without sacrificing the signal (Fig. 1a). Although the astigmatic PSF is also affected by field-dependent aberrations, the depth calculation can be optimized by dividing the entire FOV into different sub-regions with a localized look-up table for each region (Fig. 1c).

From a broader perspective, other techniques are emerging that can achieve 3D imaging of microcirculation over large (centimeter-scale) FOV. For instance, optoacoustic tomography (OAT) attains large-scale volumetric imaging with high temporal resolution by detecting high-frequency acoustic vibrations generated by absorption of short light pulses[39,40]. The spatial resolution of OAT is determined by acoustic diffraction (typically in the range of hundreds of micrometers), which is insufficient for resolving fine structures on a capillary scale. On the other hand, optoacoustic microscopy (OAM) employing focused optical beams can achieve high spatial resolution equivalent to optical microscopy at the detriment of poor temporal resolution limited by the typical point-by-point scanning implementations[41]. In general, optoacoustic imaging employs sophisticated pulsed nanosecond laser sources making it less affordable in comparison to fluorescence-based methods. The emerging ultrasound localization microscopy (ULM) techniques that employ localization of single microbubble or nanodroplet scatterers have similarly achieved super-resolution imaging in the ~1/10 of ultrasound wavelength range and deep penetration into the rodent brain by capitalizing on the insignificant scattering of ultrasound waves compared to photons in soft biological tissues[42–44]. However, as compared to the suggested optical localization approaches, the ULM offers inferior spatial resolution, which is further compromised by severe skull-induced acoustic aberrations when imaging transcranially. Furthermore, similarly to the SL case, a direct tradeoff between image quality and frame rate exits for ULM, which is averted by the MI method. Since ULM typically relies on ultrasound beamforming to select single/several coronal slices at a time, the ability to attain real-time imaging across a major part of the cortex in 3D is limited with this method. The ability to rapidly capture activity over large portion of the cortex in a transversal view is of key importance for some applications involving monitoring of large-scale brain dynamics, such as collateral recruitment post ischemic stroke as demonstrated in this work.

We applied the newly developed volumetric wide-field fluorescence microscopy technique to visualize ischemic stroke events with high resolution over large FOV while simultaneously providing quantitative perfusion information across both brain hemispheres unattainable by other optical modalities. Collateral recruitment, serving as an alternative blood supply to the infarct region, is of great interest in stroke as it crucially determines infarct severity, treatment efficacy and recovery[27]. However, collateral vessels is a challenging target for the conventional fluorescence microscopy techniques like 2PM, which typically involve highly invasive craniotomy procedures with FOV restricted to a few hundreds of micrometers thus lacking global functional information[45]. Planar (2D) speckle-based methods like LSCI are unable to discriminate single collateral vessels due to poor spatial resolution inflicted by light scattering[31]. Instead, Doppler optical coherence tomography extracts blood flow information while accurate velocity estimation relies on dense sampling of A scans and priori Doppler angle[46]. In our experiments, the large-scale, transcranial high temporal resolution MI method was able to reveal the collateral recruitment phenomenon and its unique perfusion patterns in exquisite detail.

In conclusion, we proposed a wide-field volumetric fluorescence microscopy technique based on astigmatic PSF and demonstrated its application for transcranial mapping of the murine cortical microcirculation. By utilizing two alternative fluorescence source localization approaches, we performed real-time 3D large-scale imaging of the brain vascular network at capillary level resolution while further uncovering functional information pertaining direction and velocity of the cerebral blood flow. This has allowed the observation of previously undocumented concurrent collateral recruitment events occurring on a sub-second scale with full-field functional information in a murine model of an acute ischemic stroke. The reported technique offers a wealth of possibilities for non- or minimally invasive imaging of large-scale biodynamics with an unmatched combination of high spatio-temporal resolution and large field of view.

## Methods

### The volumetric wide-field fluorescence microscopy setup

We propose two possible implementations for astigmatism-based 3D wide-field fluorescence microscopy using either multifocal illumination (MI) or sparse localization (SL) of point emitters. In fact, the two approaches can be seamlessly integrated in one system where they share the same detection path while differing in the illumination path (Fig. 1a). Continuous wave (CW) laser sources at 473 nm (FPYL-473-1000-LED, Frankfurt Laser Company, Germany) or 660 nm (gem 660–500 mW, Laser Quantum, USA) wavelength were used for fluorescence excitation based on the selected fluorophores. For the SL approach, the laser beam was coupled into a commercial fiber bundle to provide epi-illumination. In the case of MI, the beam was first expanded with a 4-fold beam expander consisting of two achromatic doublets (ACN254-050-A and AC254-200-A, Thorlabs, USA). The collimated beam was subsequently directed to a half-wave plate (AHWP10M-600, Thorlabs, USA) to adjust the polarization orientation before it enters the two-dimensional acousto-optical deflector (2D AOD, DTSXY-400-532, AA Opto-Electronic, France), which provides a scanning rate up to 95 kHz with 15-bit angle accuracy within 2.29° angle range. After passing through 2D AOD, the beam was split into $17 \times 17$ beamlets with a 0.3° inter-angle by a beam-splitting grating (DE-R 243, Holoeye Photonics AG, Germany). These beamlets were reflected by dichroic mirror (F38-663, Semrock, USA) and focused by an objective (CLS-SL, EFL = 70 mm, Thorlabs, USA) to project the multifocal illumination pattern on the sample over a FOV of $5.6 \times 5.6$ mm$^2$. For the light detection path, the backscattered fluorescence was collected with the same objective, focused with a tube lens (AF micro-Nikkor 105 mm, Nikon, Japan) and filtered with a long-pass filter (FGL695, Thorlabs, USA) before entering the sensor plane of the CMOS camera (pco.dimax S1, PCO AG, Germany). The high-speed camera features a fast frame rate up to 4.4 kHz at full pixel resolution of $1008 \times 1008$ pixels. Image acquisition was performed with Camware software (version 4.12, PCO AG, Germany). A weakly focusing cylindrical lens with 8200 mm focal length was inserted between the objective and the tube lens to introduce astigmatism. Synchronization of the AOD and camera was achieved with an external trigger signal from a digital I/O card (NI-PCIe 6536b, National Instrument, USA) to realize fast 3D imaging across the FOV.

### System characterization and depth calibration

To characterize lateral resolution of the imaging system, orange fluorescent beads (460/594 nm, 1–5 μm, Cospheric, USA) were randomly distributed on a glass coverslip and imaged with MI under 473 nm excitation using $120 \times 120$ scanning steps with 3.3 μm step size according to Nyquist-Shannon sampling theorem. The intensity profile of a randomly-chosen bead was extracted and fitted with Gaussian function. The lateral resolution was determined by the full-width-at-half-maximum (FWHM) of fitted curve. In order to calibrate the projection between the astigmatic PSF and depth (i.e., the z coordinate),

an image stack of the fluorescent slide was acquired with the z-position controlled with a commercial motorized stage (TDC001, Thorlabs, USA) with a step size of 50 μm. Subsequently, the link between the astigmatic PSF and z coordinates was established.

## Image reconstruction

For rendering the SL microscopy images, each captured frame was first enhanced by subtracting the mean intensity (background) of the entire raw image stack. The lateral positions of fluorescent emitters in each frame were localized by calculating the centroid of connected-component regions after using an adaptive threshold. Each centered subregion was extracted from the raw image, followed by calculating the axis length of PSF via iterative 2D elliptical Gaussian fitting[22]:

$$f(x,y) = a \times \exp\left(-2 \times \frac{(x-x_0)^2}{w_x^2} - 2 \times \frac{(y-y_0)^2}{w_y^2}\right) + c \qquad (1)$$

where $a$ is the height; $x$ and $y$ are the lateral coordinates; $x_0$ and $y_0$ are the center positions; $w_x$ and $w_y$ stand for the widths of the PSF, and $c$ corresponds to the noise level. The axial position was estimated according to the ellipticity of PSF and the depth calibration curve with the ellipticity defined via[47,48]

$$ellipticity = \frac{w_x - w_y}{w_x + w_y} \qquad (2)$$

In super-resolution microscopy, the length of both PSF axes is commonly employed to search for the best matched depth[22]. Here the PSF ellipticity was instead chosen for depth retrieval because in this case the calibration curve manifested higher fidelity under different scattering conditions as compared to estimations based on single-axis length or difference between both axes of the astigmatic PSF (Supplementary Fig. 3). The Simpletracker algorithm[49] was then applied to recognize the same bead in consecutive frames with the trajectories built based on their positions. The parameters for Simpletracker were chosen according to the actual frame rate to mitigate false trajectories due to excessive density of the flowing fluorescent beads. At the same time, the flow velocity at each pixel was calculated according to the frame rate and spatial displacement. The 3D localization image was finally rendered by superposing all the 3D trajectories within the FOV. Similarly, flow velocity map was reconstructed by superimposing the averaged velocity at each pixel.

Image reconstruction with MI includes three steps. The first step is localization of the multi-focal illumination pattern in each frame. Since the fluorophore distribution across the imaged FOV may not be uniform, some illumination spots could be missing in the raw image if localization is performed on the single spot level. Instead, all the scanning positions of the illumination pattern were recorded with a uniformly fluorescent slide and used as reference for the signal extraction. Subsequently, the position of each illumination spot in the recorded frames can be localized by searching for local maxima with an adaptive threshold around the pre-recorded position, followed by the extraction of centered subregions. To minimize the field dependent errors, each raw image is divided into 25 subregions with known illumination coordinates. In the second step, the ellipticity of PSF is calculated with the axial position value assigned according to the previously established depth calibration file. In the third step, signal intensity of each spot in each scanning frame is retrieved by applying a digital pinhole to reject cross-talk between adjacent emitters and out-of-focus light. The final compounded 3D images are then obtained by superimposing all the filtered frames scanned with the AOD. All the data analyses were performed with customized MATLAB (MathWorks, MATLAB R2019b, USA) programs.

## Vessel segmentation and analysis

The previously reported vessel segmentation and analysis algorithm named PostProGUI[50] was employed to quantify vascular network changes on the contralateral and ipsilateral side of the mouse brain post ischemic stroke (supplementary Fig. 4). Three ROI pairs were selected from the MIP of reconstructed 3D brain image with symmetrical positions in two hemispheres. To avoid manual bias, all the ROIs underwent the fixed pre-processing pipeline including 2-fold linear interpolation, image-guided filtering and Frangi filtering for contrast enhancement. Subsequently, image binarization was realized with automatic threshold based on Otsu's algorithm, followed by vessel skeletonization procedure[51]. Specifically, morphological thinning was applied to the segmented binary image to compute the vessel centerlines, rendering another binarized image. Subsequently, branch points that connect different vessel branches were recognized whenever more than two non-zero neighboring pixels were detected. The removal of such branch points automatically divided the connected vessels into vessel branches. After the identification of vessel branches, advanced morphological parameters such as fill fraction (the total blood vessel area divided by the total imaged area) and vessel branch number were calculated for further statistical analysis.

## Laser speckle contrast imaging

A commercial laser speckle contrast imaging (LSCI) system (FLPI, Moor Instrument, UK) was used to monitor the cortical perfusion and to verify the blood flow reduction after stroke. The LSCI images were generated with arbitrary units in a 32-color palette by the MoorFLPI software (version 4.0, Moor Instrument, UK), then further co-registered with the reconstructed image collected by MI method using manually chosen features and *imregdemons* function on MATLAB.

## Scanning two-photon microscopy

Two-photon imaging was performed with a custom-built microscope[52] equipped with a tunable femtosecond laser (Chameleon Discovery NX, Coherent Inc, USA). A 16× water-immersion objective (CFI75 LWD 16X W, NA = 0.8, Nikon, Japan) was used for the in vivo validation. Synchronization of galvo mirror and fluorescence collection was achieved with ScanImage (r3.8.1, Janelia Research Campus)[53]. A piezo motor-driven linear stage was used for z-scanning across the sample. Back-scattered fluorescence photons passing through a bandpass filter (FF01-475/64-25, Semrock, USA) were then collected with a photo-multiplier tube (H9305-03, Hamamatsu, Japan). For velocity measurement, line scanning parallel to flow direction was performed to generate kymographs. The velocity of red blood cells was estimated based on radon transform[54,55].

## Phantom preparation

Two phantoms were prepared to evaluate the depth estimation accuracy of the MI method under 660 nm excitation. A tilted microscope slide with a thin layer of Cy5.5 dye was raster scanned over $45 \times 45$ positions with the AOD. Another phantom was comprised of two single-channel microfluidic chips (inner diameter = 50 μm, Microfluidic ChipShop, Germany) overlaid in a crossing orientation and separated by approximately 260 μm along the axial (depth) direction. Similarly, the illumination pattern was raster scanned with $45 \times 45$ positions and the 3D compounded image was reconstructed as described in the image reconstruction section.

For validating the depth estimation by the SL method, orange fluorescence 1–5 μm diameter beads (460/594 nm, Cospheric, USA) were mixed with UV-curing glue (NOA61, Thorlabs, USA) and poured onto the microscope slide resulting in their effective distribution across different depths. The phantom was subsequently exposed to UV light for 10 min. SL image stack was collected by translating the

phantom along the *z* axis with 10 µm step size using a motorized stage (MLJ150, Thorlabs, USA).

### In vivo animal experiments

Two athymic nude-Foxn1nu mice (Envigo BMS B.V., Netherlands) were used for in vivo imaging of the healthy cerebral microcirculation. The mice were anesthetized with isoflurane (5% for induction and 1.5% for maintenance) in a mixture of oxygen and medical air with flow rates of 0.2 L/min and 0.8 L/min, respectively. The scalp of both mice was removed to reduce severe light scattering from skin after subcutaneous injection of analgesics (Buprenorphine, 0.1 mg/kg). For the SL microscopy, 100 µL solution of the same orange fluorescent beads in phosphate-buffered-saline used in phantom experiments were slowly injected into tail vein of a nude mouse (12 weeks old, female) followed by wide-field recording with a frame rate of 200 Hz for 3.5 min. For the MI microscopy, a nude mouse (8 weeks old, female) was injected instead with 50 µL (2 mg/ml concentration) of Cy5.5 fluorescent dye. The images were then collected at 2.25 kHz AOD scanning frequency and camera acquisition rate, corresponding to 10 Hz effective compounded frame rate when using $15 \times 15$ scanning positions. For stroke study, focal cerebral ischemia was induced[56]. One female and two male C57BL/6J mice (6–7 weeks old, Envigo BMS B.V., Netherlands) were anesthetized intraperitoneally (i.p.) with a mixture of fentanyl (0.05 mg/kg bodyweight; Sintenyl, Sintetica), midazolam (5 mg/kg bodyweight; Dormicum, Roche), and medetomidine (0.5 mg/kg bodyweight; Domitor, Orion Pharma). Under anesthesia, the mouse was fixed in a stereotactic frame with body temperature kept at 37 °C using a feedback-regulated heating system throughout the surgical procedure. After scalp removal, a glass micropipette (calibrated at 15 mm/µL; Assistent ref. 555/5; Hoechst, Sondheim-Rhoen, Germany) was inserted into the lumen of the MCA followed by purified human alpha-thrombin injection (1 µL, HCT-0020, Haematologic Technologies Inc., USA) to induce MCA occlusion. The pipette was removed after a stable clot was formed. The mouse was subsequently imaged with LSCI and MI microscopy, respectively.

A C57BL/6J mouse (10 weeks old, female, Envigo BMS B.V., Netherlands) was used for blood velocity cross-validation against 2PM. Animals were anesthetized intraperitoneally with a mixture of fentanyl (0.05 mg/kg bodyweight), midazolam (5 mg/kg bodyweight), and medetomidine (0.5 mg/kg bodyweight). A 3.5 mm diameter craniotomy was performed over the primary somatosensory cortex with a dental drill (Bien-Air) followed by implantation of $3 \times 3$ mm glass coverslip and a titanium head plate. For imaging session, anesthesia was induced with isoflurane with the same parameters mentioned above. An anatomical image stack of the mouse brain vessels was at first collected with 2PM at 800 nm after the intravenous injection of 100 µL cascade blue (2.5% v/v, D1976, Thermofisher, USA). Velocity measurement was performed over three vessel branches, selected based on the angiographic image. The same mouse was subsequently imaged with the SL technique post fluorescent beads injection. After the experiments, all mice were euthanized while still under deep anesthesia.

Animals were housed in ventilated cages inside a temperature-controlled room under a 12 h dark/light cycle. The temperature was kept at 22 °C and relative humidity was kept at 50%. Pelleted food (3437PXL15, Cargill) and water were provided. All animal experiments were performed in accordance with the Swiss Federal Act on Animal Protection and approved by the Cantonal Veterinary Office Zurich (permits ZH165/2019 and ZH161/2018).

### Reporting summary

Further information on research design is available in the Nature Portfolio Reporting Summary linked to this article.

## Data availability

The main data supporting the finding of this study are available within the main text or supplementary information. Raw data for Figs. 1b, c, 2c, j, 4e, g and supplementary Figs. 1b, 2d, 3a, c are provided in the Source Data file. The raw datasets before image reconstruction are too large to be publicly shared, yet they are available for research purposes from the corresponding author upon request. Source data are provided with this paper.

## Code availability

The code that supports the findings of this study is available for research purposes from the corresponding author upon request.

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

## Acknowledgements

D.R. acknowledges funding from the US National Institutes of Health (UF1-NS107680 and R01-NS126102) and the Swiss National Science Foundation (310030_192757). S.W. acknowledges funding from the Swiss National Science Foundation (310030_192757, PP00P3_170683), and the UZH CRPP Stroke.

## Author contributions

Q.Z. and Z.C. conceived the experimental system and carried out the experiments. Q.Z., Z.C. and Y.L. conducted the data analysis and visualization. M.E.A., C.G., J.D. and M.R. contributed to the animal experiments and interpretation of the results. B.W. and S.W. provided the stroke model and guided the animal experiments. D.R. was involved in experimental design and planning and supervised the work. All authors contributed to writing and revising the manuscript.

## Competing interests

The authors declare no competing interest.
