## [Peer Review File · Nature Communications]

Three-dimensional wide-field fluorescence microscopy for transcranial mapping of cortical microcirculationEditorial Note: This manuscript has been previously reviewed at another journal that is not operating a transparent peer review scheme. This document only contains reviewer comments and rebuttal letters for versions considered at Nature Communications.

REVIEWERS' COMMENTS

Reviewer #1 (Remarks to the Author):

In this manuscript the Authors describe their development and testing of an optical microscopy technique able to acquire volumetric data with high temporal resolution and with a capillary-level spatial resolution. I reviewed the previous version of this manuscript which was originally submitted to [redacted].

I confirm my original positive assessment of the manuscript: it is well written, the results are sound, the existing literature is appropriately referenced and the topic is of interest for the optical imaging community.

A major concern in the original submission was a potentially inadequate impact of the manuscript for a high impact-factor journal such as [redacted]. This issue has been addressed in this new submission to Nature Communications.

In the original submission I pointed out several other issues in the manuscript. They were all convincingly addressed by the Authors in their rebuttal and in the revised version of the manuscript.

There is only one remaining issue in the present version of the manuscript. I pointed out a weakness point in the original version of the manuscript: the limitedness of the demonstrated application field. The presented method is in principle applicable to a diverse range of samples, provided that the process or the structure under investigation is characterized by either spatial and/or temporal sparsity. On the other hand, in the manuscript the Authors report its application uniquely to the study of microcirculation in the murine brain. This issue was resolved by the Authors in the revised version of the main text of the manuscript by expanding the Discussion and Conclusion sections to emphasize the currently known limitations of the method and potential future application directions. Nevertheless, it seems to me that the title of the manuscript ("Volumetric wide-field fluorescence microscopy") is too general, potentially anticipating to the reader that a multiplicity of samples and applications would be detailed in the article.

Consequently, I recommend modifying the title to reflect this limitation. Among the possible titles, I would suggest: "Volumetric wide-field fluorescence microscopy, applied to the imaging of cortical microcirculation".

Apart from this minor issue, I recommend this manuscript for publication.

Reviewer #3 (Remarks to the Author):

Overall, I think the authors have undertaken major efforts to address my concerns and revise their manuscript. I have no other comments, and I think the paper is suitable for publication in Nature Communications.

Point-by-Point Response to the Reviewers' comments

Reviewer #1

In this manuscript the Authors describe their development and testing of an optical microscopy technique able to acquire volumetric data with high temporal resolution and with a capillary-level spatial resolution. I reviewed the previous version of this manuscript which was originally submitted to [redacted]. I confirm my original positive assessment of the manuscript: it is well written, the results are sound, the existing literature is appropriately referenced and the topic is of interest for the optical imaging community.

A major concern in the original submission was a potentially inadequate impact of the manuscript for a high impact-factor journal such as [redacted]. This issue has been addressed in this new submission to Nature Communications. In the original submission I pointed out several other issues in the manuscript. They were all convincingly addressed by the Authors in their rebuttal and in the revised version of the manuscript.

There is only one remaining issue in the present version of the manuscript. I pointed out a weakness point in the original version of the manuscript: the limitedness of the demonstrated application field. The presented method is in principle applicable to a diverse range of samples, provided that the process or the structure under investigation is characterized by either spatial and/or temporal sparsity. On the other hand, in the manuscript the Authors report its application uniquely to the study of microcirculation in the murine brain. This issue was resolved by the Authors in the revised version of the main text of the manuscript by expanding the Discussion and Conclusion sections to emphasize the currently known limitations of the method and potential future application directions. Nevertheless, it seems to me that the title of the manuscript ("Volumetric wide-field fluorescence microscopy") is too general, potentially anticipating to the reader that a multiplicity of samples and applications would be detailed in the article. Consequently, I recommend modifying the title to reflect this limitation. Among the possible titles, I would suggest: "Volumetric wide-field fluorescence microscopy, applied to the imaging of cortical microcirculation". Apart from this minor issue, I recommend this manuscript for publication.

Reply: We thank the Reviewer for the positive and constructive comments on our work. The title of the manuscript has been amended to "Three-dimensional wide-field fluorescence microscopy for transcranial mapping of cortical microcirculation". We want to express our gratitude for the precious revision suggestions in both rounds which are very helpful to improve the quality of this work.

Reviewer #3

Overall, I think the authors have undertaken major efforts to address my concerns and revise their manuscript. I have no other comments, and I think the paper is suitable for publication in Nature Communications.

Reply: We thank the Reviewer for the positive feedback to our revision and we are incredibly grateful for the constructive revision suggestions which are very helpful to improve the quality of this work.